# Trends in flood losses in Europe over the past 150 years

Dominik Paprotny [1,2], Antonia Sebastian[1,3], Oswaldo Morales-Nápoles [1] & Sebastiaan N. Jonkman[1]

Adverse consequences of floods change in time and are influenced by both natural and socio-economic trends and interactions. In Europe, previous studies of historical flood losses corrected for demographic and economic growth ('normalized') have been limited in temporal and spatial extent, leading to an incomplete representation of trends in losses over time. Here we utilize a gridded reconstruction of flood exposure in 37 European countries and a new database of damaging floods since 1870. Our results indicate that, after correcting for changes in flood exposure, there has been an increase in annually inundated area and number of persons affected since 1870, contrasted by a substantial decrease in flood fatalities. For more recent decades we also found a considerable decline in financial losses per year. We estimate, however, that there is large underreporting of smaller floods beyond most recent years, and show that underreporting has a substantial impact on observed trends.

[1] Department of Hydraulic Engineering, Faculty of Civil Engineering and Geosciences, Delft University of Technology, Stevinweg 1, 2628 CNDelft, The Netherlands. [2] Disaster Risk Management, Directorate E–Space, Security and Migration, European Commission, Joint Research Centre (JRC), via E. Fermi 2749, I-21027 Ispra, VA, Italy. [3] Department of Civil and Environmental Engineering, Rice University, 6100 Main Street, Houston, TX 77005, USA. Correspondence and requests for materials should be addressed to D.P. (email: d.paprotny@tudelft.nl)

Extreme hydrological events are generally predicted to become more frequent and damaging in Europe due to warming climate[1–7]. Though consensus seems to exist regarding the trajectory of future climatic developments seem certain, there is less confidence in the changes in flood losses as a result of climate change so far[8–11]. Qualitative and quantitative hydrological studies for Europe have indicated no general continental-wide trend in river flood occurrences, extreme precipitation, or annual maxima of runoff[12–14]. However, substantial variations between different catchments have been observed, ranging from an increase in north-western Europe to no trend or a decrease in other parts of the continent[15, 16]. Similar findings were reported for storminess along the European coasts[17, 18].

Natural hazards are phenomena that inherently involve adverse consequences to society. Therefore, analyses of long-term trends in flood losses should also account for changes in size and distribution of population and assets[19, 20]. Without correcting reported losses for spatial and temporal changes in exposure, previous studies report a significant upward trend in losses[21–23]. However, after adjusting nominal losses for demographic and economic growth, no significant trends in flood losses, both on European scale[14, 24] and for individual countries were observed[25–27]. Such 'normalization' processes have also proven to be important for explaining trends in other natural hazards[28–30].

Still, there are two main limitations in existing analyses. First, historical disaster loss data are not temporally homogenous, with the number of flood events for which quantitative information is available declining quickly when moving back in time. The starting point for many studies is in the vicinity of the year 1970 or later[31, 32]. International databases of natural hazards (EM-DAT[33], NatCatService[34], Dartmouth Flood Observatory[35] or European Environment Agency[36]) provide reasonable coverage only beginning with the 1980s. Comprehensive and publicly available national repositories of disaster loss data are few in Europe and, those that are available, focus on flood and landslide events[37–41]. Moreover, the completeness and extent of information contained in existing data sets varies to a significant degree. In effect, large-scale studies usually rely entirely on the contents of global or continental databases, while national studies are shaped by the specifics of locally available data. This leads to considerable uncertainties when examining trends at the continental scale or comparing trends between countries.

Second, in virtually all studies, socio-economic variables are considered at the national level; only Munich Re utilized a coarse 1°×1° grid of exposure data (approx. 5000–9000 km$^2$ over Europe)[24]. High resolution is of particular importance for analyzing flood exposure, which is relatively limited in space: at present time less than 10% of European territory is at risk of river or coastal flooding[42]. A few national studies that have analyzed changes in exposure found different trends in population or housing stock inside and outside hazard zones[43–46], which shows the importance of using a sufficient resolution of the analysis. Furthermore, trends in exposure and normalization of reported losses have been carried out with many different economic variables depending on the study, such as gross domestic product (GDP), variously defined wealth or housing stock.

In this study, we address the aforementioned limitations (short time series and low spatial resolution) of previous assessments of flood trends for Europe using two datasets which constitute a new publicly available database 'Historical Analysis of Natural Hazards in Europe' (HANZE)[47, 48]. The first dataset (HANZE-Exposure) contains high-resolution (100 m) maps of land cover/use, population, GDP and wealth in 37 European countries and territories from 1870 to 2020. The maps were created by estimating changes in the distribution of land cover/use and population relative to the year 2011, for which detailed gridded datasets are available[49, 50] (see Methods section for more details). The second dataset (HANZE-Events) includes records of 1,564 damaging flood events that occurred within the same domain between 1870 and 2016 and had adverse consequences to people or property (damaging floods). Our results indicate an increase in inundated area and the number of persons affected contrasted by a consistent decline in flood fatalities for the period 1870–2016, with no significant trend in (normalized) financial losses. However, for the period after 1950, we observe a considerable decline in fatalities and (normalized) monetary losses. Moreover, we show that when correcting for underreporting, the annual number of flood events and persons affected have increased much less than calculated using uncorrected series (and possibly declined since the mid-20th century), and that financial losses have declined over time. We foresee numerous applications of the HANZE database for further studies, including an analysis of trends for other hazards, an assessment of the potential impacts of climate change on historical losses, and studies of individual events and their impact on flood management.

## Results

**Trends in exposure.** Between 1870 and 2016, Europe experienced substantial growth in population (130%), urban area (more than 1000%), and wealth (more than 2000% constant prices). However, there has been large variability in patterns of development between regions. In 8% of European regions (NUTS 3), the total population in 2016 was lower than in 1870. Rural population across the continent declined, and fixed assets in agriculture barely changed in contrast with large increases in wealth in the housing, industry and services sectors (Supplementary Fig. 1). Most important for this study are relative trends within and outside of flood-prone areas. Since 1870, the percentage of population, GDP and wealth exposed to the 100-year flood has decreased slightly for river floods, but increased for coastal floods (Fig. 1). When analyzed at the continental scale, those trends are partly caused by the aforementioned rates of demographic and economic growth between regions (Supplementary Fig. 2). As the map in Fig. 1 shows, while overall exposure to floods has declined in most countries, especially those in central and northern Europe, relative exposure has increased in several western and southern European states including France, Germany, Italy and the Netherlands. In general, changes in exposure of production (measured by GDP) and wealth are in line with trends in population, with some exceptions, e.g., in Italy and Hungary, where the percentage of wealth exposed has not changed since 1870 despite growth in the relative exposure of their national populations.

**Distribution of flood events in Europe.** The HANZE database includes records for 1564 flood events (1870–2016), of which 879 (56%) are flash floods, i.e., river floods lasting less than 24 h, 606 (39%) are river floods, 56 (4%) are coastal floods and the remaining 23 (1.5%) are compound events, i.e., floods caused by a co-occurrence of storm surge and high river flows. For the purpose of this analysis, 'flood events' (or simply 'events') refer only to damaging floods fulfilling criteria for inclusion in the HANZE database (see Methods for details). Flood events are very unevenly distributed, both during any given year and geographically (Fig. 2). In southern Europe, flash floods constituted the majority of flood events, and were most prevalent between September and November. In central and western Europe, river floods were more frequent than flash floods, with flood losses concentrated between June and August. In northern Europe, floods were mostly caused by snowmelt and rarely resulted in significant losses. Coastal

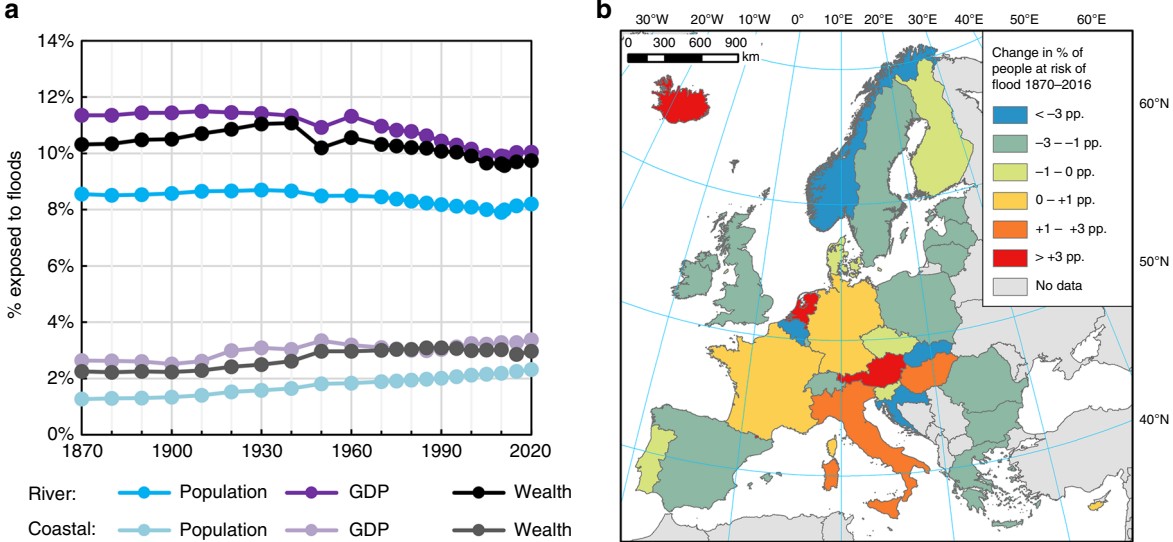

**Fig. 1** Trends in flood exposure. Percent of the population exposed to the 100-year river and coastal flood in Europe (**a**), including short-term projection to year 2020, and change in population exposed (**b**), in percentage points, to the 100-year flood (either river or coastal) in each country (1870–2016). Source of data: HANZE database[47] with country borders from PBL[82]

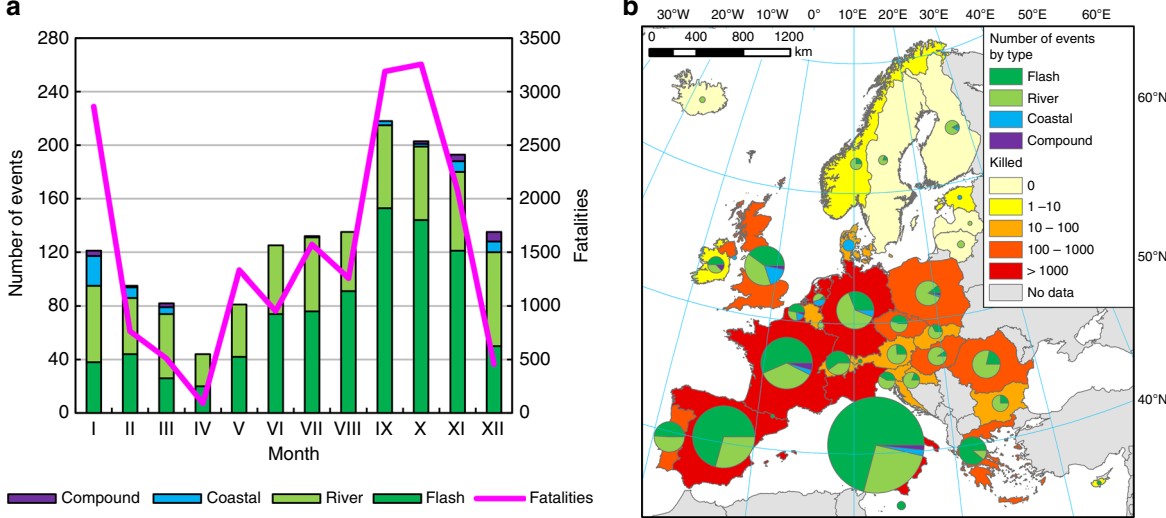

**Fig. 2** Flood occurrences and fatalities. Total number of flood events and fatalities (unadjusted, reported values) between 1870 and 2016, **a** by month and **b** by country. Source of data: HANZE database[47] with country borders from PBL[82]

floods were mostly recorded in regions which border the North and Baltic seas.

In total, HANZE contains information on flood events that affected 1005 regions, or 74% of all NUTS3 regions within the study area. The number of floods by region is presented in Supplemental Fig. 3. On average, a flood event affected 2.8 NUTS 3 regions. The spatial distribution of floods contained in the database is heavily influenced by availability of historical records. More than half of the events in the database occurred in only three countries, namely Italy (36%), Spain (15%) and France (10%), all of which have publicly available and searchable databases of historical floods. Thus, the large number of recorded flood events in those countries is a result of better coverage of events with relatively small impact on population or assets. In contrast, total flood losses are more evenly spread out across Europe and less than a third of people affected by floods resided in the aforementioned three countries. This is partially a result of better coverage of major flood events across all countries, whereas

flood events recorded in Italy, Spain and France were dominated by flash floods.

It should be noted that quantitative information on floods losses was not always obtainable. The most frequently available statistic was the number of fatalities, as they were recorded for 1547 flood events (99%), of which 372 events resulted in no deaths. For the remaining 17 events some fatalities were reported to have occurred, but the exact number of deaths was unknown. Information on the total flooded area was only available for 157 events (10%), persons affected for 682 (44%) and monetary losses for 560 (36%).

**Trends in reported and normalized flood losses.** In Fig. 3, the records from the database are aggregated per year, and shown in two variants. In saturated colors, the original, unadjusted values of damages are shown as reported in historical records. Only the monetary value of losses was adjusted for price of inflation and converted to 2011 euros. In less intense colors, the normalized

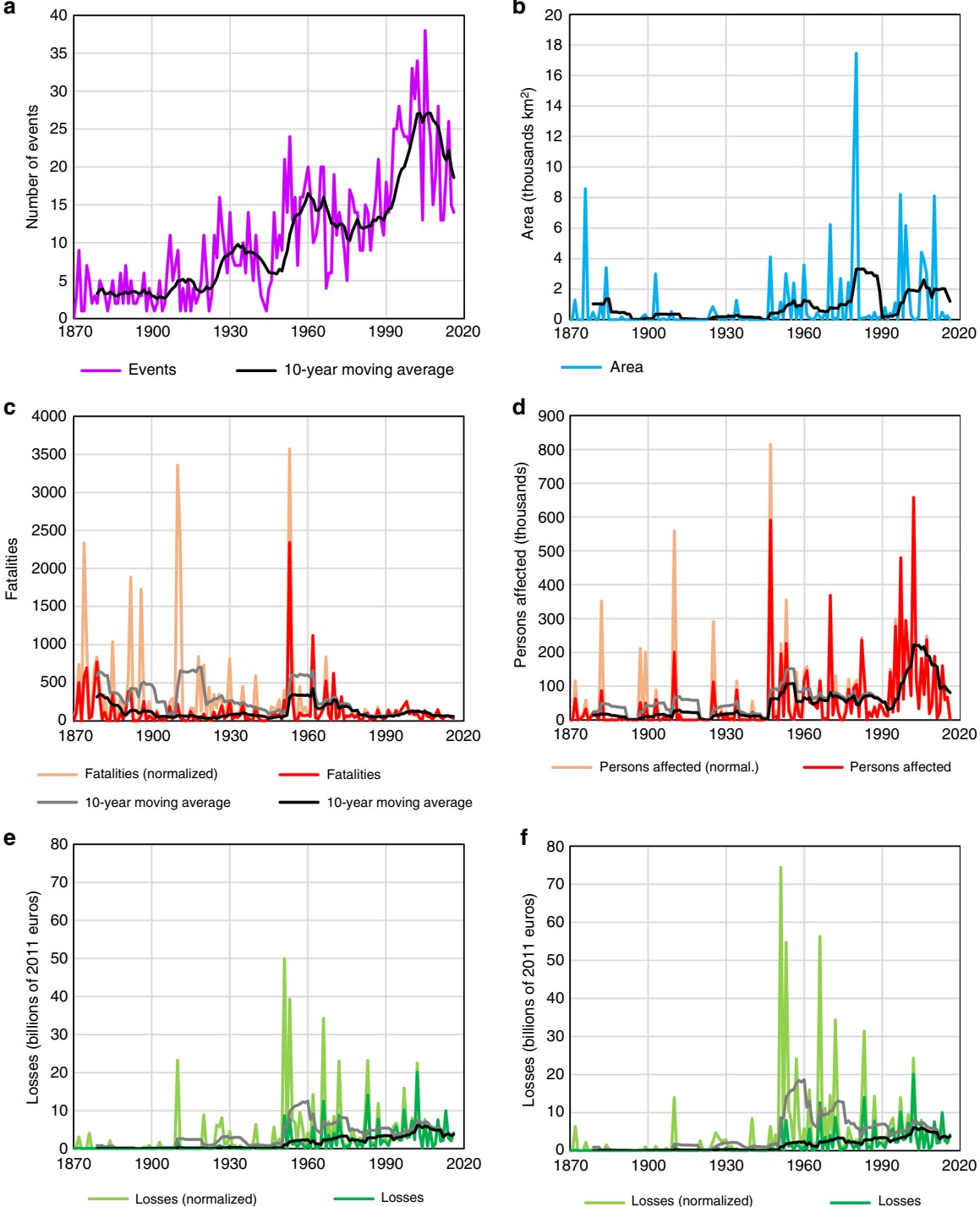

**Fig. 3** Trends in flood losses per year. Comparison of unadjusted, reported values (dark colors) and normalized values, i.e., adjusted to 2011 levels of exposure (lighter colors), for **a** number of events; **b** area inundated; **c** fatalities; **d** persons affected; **e** financial value of losses with normalization by GDP; and **f** financial value of losses with normalization by wealth

values, i.e., those adjusted for change in population, GDP or wealth within the individual floods' footprints, are presented between the year of the event and 2011. It is important to note that vulnerability to floods is assumed to be constant and that the reported losses are only multiplied by the change in number of persons, production or assets in a given footprint (see Methods section for details).

The resulting trends are reported in Table 1 for five periods: 1870–2016, 1900–2016, 1930–2016, 1950–2016, and 1970–2016.

Most flood events recorded in the database occurred in recent decades, with relatively small numbers of events reported for the late 19th century. Over most of the period of record, the total area inundated increased significantly, however no significant trend is observed after 1930. Given that area flooded is known only for a tenth of all events in the database, confidence in this finding is low. In contrast, the number of fatalities is available for almost all flood events in the database and a negative trend of at least 1% per year is observed, even though it is only statistically significant

**Table 1 Comparison of trends in annual flood losses**

| Starting year | Reported | | | | | Normalized | | | | Normalized and gap-filled | | | | |
|---|---|---|---|---|---|---|---|---|---|---|---|---|---|---|
| | Events | Area | Fatalities | Affected | Losses | Fatalities | Affected | Losses[a] | Losses[b] | Area | Fatalities | Affected | Losses[a] | Losses[b] |
| 1870 | 1.5[c] | 1.4[c] | −0.3 | 2.0[c] | 3.0[c] | −1.1[c] | 1.1[c] | 1.5[c] | 1.4[c] | 1.6[c] | −1.2[c] | 0.7[c] | 0.2 | −0.1 |
| 1900 | 1.5[c] | 2.0[c] | 0.2 | 2.0[c] | 2.8[c] | −1.4[c] | 1.2[c] | 1.0 | 0.9 | 1.8[c] | −1.3[c] | 0.6 | 0.2 | 0.3 |
| 1930 | 1.3[c] | 1.6 | −0.9 | 1.7[c] | 2.4[c] | −1.8[c] | 1.1 | −0.1 | 0.3 | 1.7[c] | −1.8 | 0.4 | −0.5 | −0.0 |
| 1950 | 1.0[c] | 0.6 | −3.3[c] | 1.4 | 1.3 | −4.6[c] | 0.8 | −2.6[c] | −1.8 | 1.3[c] | −4.7[c] | −0.1 | −2.3[c] | −1.5[c] |
| 1970 | 1.4[c] | −1.5 | −1.7 | 1.2 | 1.3 | −1.9 | 0.9 | −1.6 | −0.6 | 1.0 | −2.0[c] | 0.3 | −1.2 | −0.3 |

Values are in % per year and equal the rate parameter in Poisson regression. The time periods all end in 2016. For uncertainty ranges, see Supplementary Figs. 5 and 6
[a]normalized by wealth
[b]normalized by GDP
[c]significant at $\alpha = 0.05$

for the period between 1950 and 2016 (see Methods section for statistical significance testing procedure). Finally, for both the number of persons affected and monetary losses adjusted for inflation, a positive trend is observed over all periods of record. However, for 1950–2016 and 1970–2016 the trend is not significant.

Normalization has a considerable effect on the observed results. The downward trend in fatalities becomes much more pronounced, reaching −4.6% per year (1950–2016). It also becomes statistically significant except for the period between 1970 and 2016; however, uncertainty regarding past exposure to floods renders the trends for this time period insignificant. Nonetheless, during the period from the 1980s to the present there have been fewer (normalized) deaths than almost any period prior. In contrast, the number of persons affected increases consistently throughout time, but the trend is less pronounced than before normalization (approximately 1% per year compared to almost 2% without adjustment). Still, the total number of flood victims peak around the year 2000. In terms of financial losses, the increase for 1870–2016 becomes smaller after normalization (1.4–1.5% per year instead of 3%), but still significant. However, when using the starting years 1900 and 1930 for the analysis, the trend in financial losses becomes statistically insignificant. The biggest shift in financial losses occurs for the period between 1950 and 2016 where the trend (−2.6% per year) is statistically significant. This is similar to the finding before normalization, however the trend is now downward rather than upward. Correcting losses by changes in both GDP and wealth indicates that losses peaked in the 1950s rather than the 2000s. In general, flood losses have been declining in the entire post-1945 period despite some noticeable cycles of higher and lower loss-generating periods.

**Trends in flood losses corrected for missing records**. Historical records of flood events often do not contain all or even most of the statistics on the consequences of floods. Hence, in order to better assess trends in flood losses, gaps in the database were filled using estimates based on an analysis of the dependence structure between all pairs of variables using copulas (see Methods). Gap-filled annual losses are presented in Fig. 4. The difference between the unadjusted and gap-filled data is clearly visible in the graphs; only in the case of the number of fatalities are the differences small. This is because there were few gaps in the historical record of the number of fatalities.

The addition of modeled data to the historical record affected many of the observed trends, both compared to reported and normalized losses (Table 1). The trend in inundated area for 1950–2016 becomes statistically significant after gap-filling (1.3% per year), while an opposite trend is indicated for 1970–2016: an annual increase of 1% (not significant) instead of an annual decrease of 1.5%. However, for the entire period 1870–2016, there is little difference in the observed upward trend after gap-filling (1.6% instead of 1.4%). In terms of the number of deaths, there is almost no change in trends, as fatalities decline across the board, with the trend for 1950–2016 reaching −4.7% per year. The number of persons affected before correcting for missing records shows an 0.8–1.2% increase across all considered time periods, while after correction, the trend decreases to at most 0.7%, annually, with a small decline during the period between 1950 and 2016. Only the 1870–2016 trend is statistically significant. Moreover, the normalized monetary value of losses after gap-filling no longer shows a significant trend for the whole period, and losses normalized by wealth increase by only 0.2% per year, while normalized by GDP decline by 0.1% per year. For all other time slices, the general trends are the same as before correcting for missing data.

**Variation in flood loss trends by area and type of flood**. Trends calculated for all events in Europe include variations within different groups of floods. Supplementary Table 2 consists of five tables identical to Table 1, but presenting the results for two subdomains: the Mediterranean countries (Cyprus, Greece, Italy, Malta, Portugal and Spain) and all other countries. The results are also shown for the different types of flood events flash floods, river floods and river/coastal/compound floods together. The tables are synthesized in Supplementary Fig. 7, in which normalized and gap-filled trends can be compared for different selections of flood events. There is a sharp contrast between the trends observed in the Mediterranean region (containing 57% of events) and all other countries. Trends for the subdomains diverge substantially over time for all variables except fatalities. Especially for the period since 1950, there are significant downward trends in the Mediterranean countries in normalized and gap-filled fatalities, persons affected and monetary losses, whereas opposite or not statistically significant trends are observed in the other parts of Europe. This difference is partly because flash floods constitute a larger share of events in the Mediterranean region than in the northern European countries. However, when looking for trends in the consequences of flash flood (56% of events) and river flood (39%) events, the differences are smaller. The decline in fatalities and number of persons affected due to flash flood events are larger than those from river floods. For economic losses, they are broadly similar.

**Estimation of underreporting of flood events**. The findings presented here include several uncertainties. One is the completeness of the database of historical floods. In principle, per each major flood event in the record, there should also be multiple smaller ones. For the purposes of this analysis, we consider flood events as 'small' or 'major' in relation to their

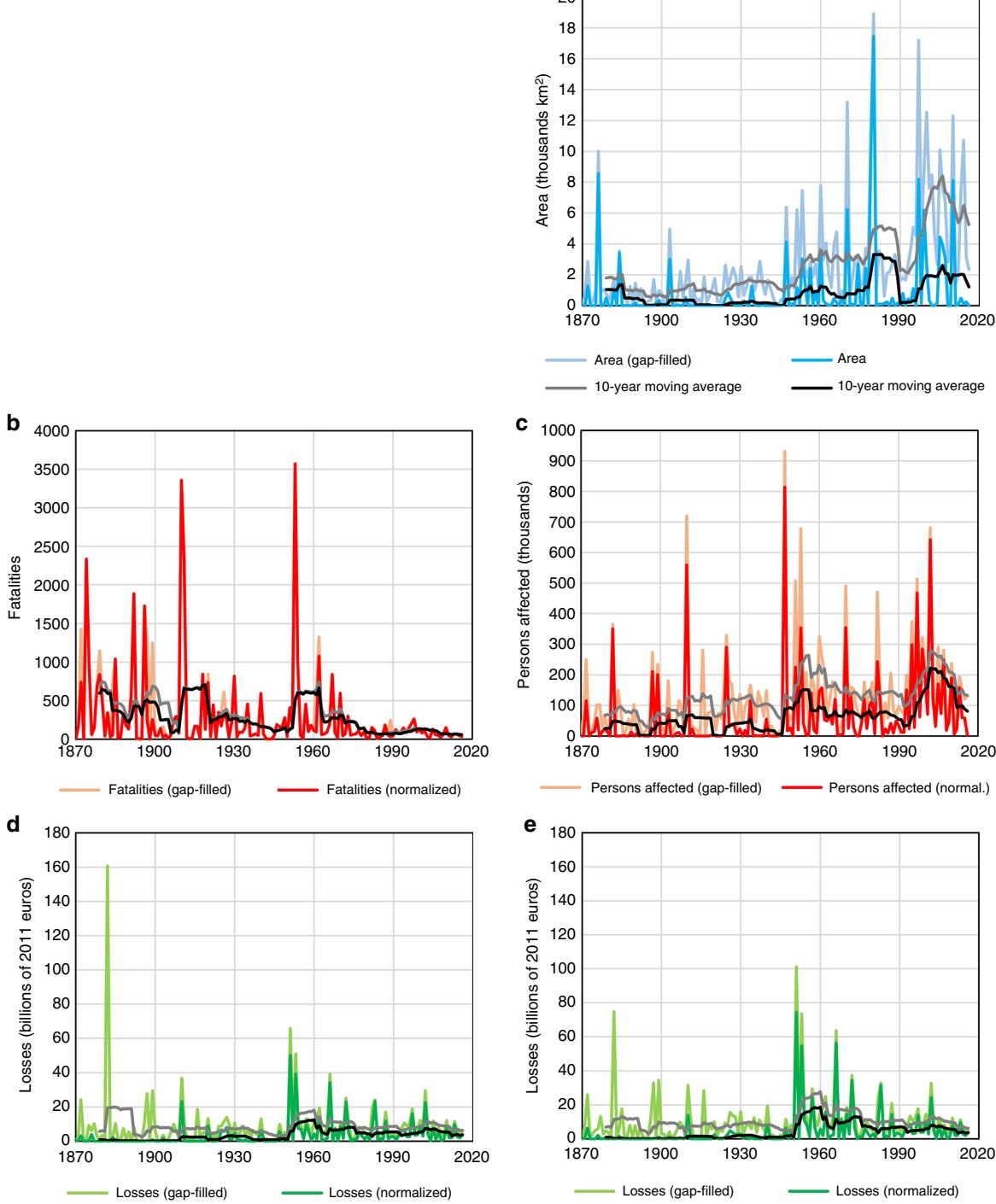

**Fig. 4** Trends in normalized flood losses per year. Comparison of losses with (lighter colors) and without gap-filling (dark colors) for **a** area inundated; **b** fatalities; **c** persons affected; **d** financial value of losses with normalization by GDP; and **e** financial value of losses with normalization by wealth

severity, i.e., the amount of losses generated by those floods relative to the overall distribution of losses for all events, where small floods are those in the lower percentiles of this distribution and major floods are those in the upper percentiles. There are relatively few small events recorded in HANZE before about 1950. If we divide the flood events by severity into quintiles (Fig. 5 and Supplementary Fig. 8), the smaller the flood, the steeper the observed trend in number of flood events. For example, the annual increase in number of flood events in the uppermost quintile (i.e., largest floods) is 0.3% per year

compared to 2% per year for those in the lowest quintile. This finding is also the same when splitting flood events by decile (with less than 0.1% increase per year in the upper 10%). This points to substantial underreporting of smaller floods historically; they are simply not mentioned in contemporary publications referring to historical events. Yet, small floods remain important since they can have a large contribution to overall damages over longer periods of time[51]. In the present, better availability of news reports and government data improves coverage considerably.

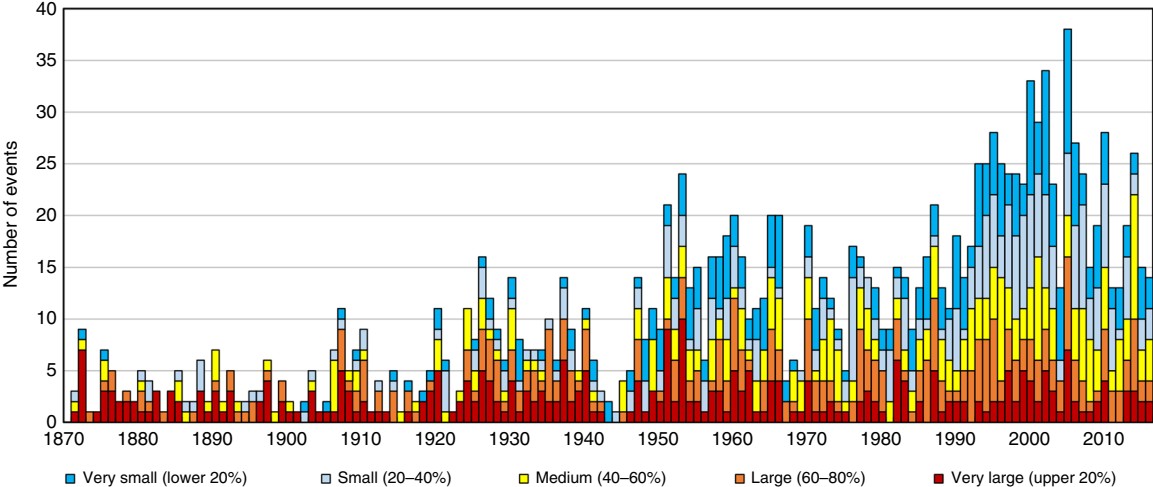

**Fig. 5** Severity of floods. Annual number of flood events classified by severity into quintiles. Classification is based on normalized and gap-filled values of losses

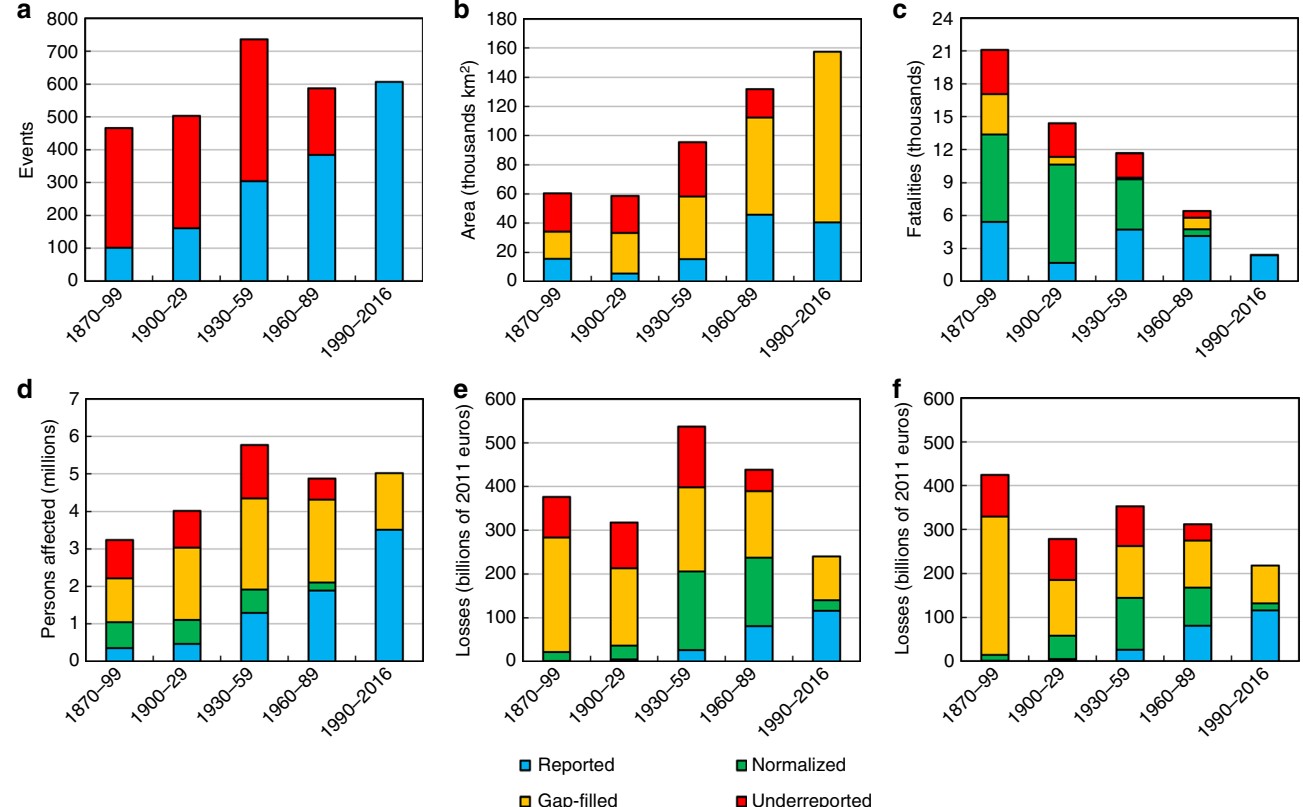

**Fig. 6** Flood losses in 30-year periods. Reported number of flood events and their consequences is summed per 30-year periods, with three types of adjustments: normalization, gap-filling of missing (normalized) loss data and estimation of underreporting of small flood events and normalized damages they caused, for **a** number of events; **b** area inundated; **c** fatalities; **d** persons affected; **e** financial value of losses with normalization by GDP; and **f** financial value of losses with normalization by wealth

To estimate the quantity of missing information, or underreporting, we adjust the number of events (except those in the upper 20%) before 1990 so that the ratio between number of events in each quintile is the same as after 1990 (see Methods for details). A summary of all adjustments to reported data is presented in Fig. 6. We find that correcting for underreporting diminishes most of the upward trend observed in the number of flood events, whereas it only slightly reduces

the growth in area inundated. Yet, given the very small number of recorded flood extents (even for the most recent events), there is considerable uncertainty in both gap-filling and the correction applied for underreporting. The decline in number of fatalities becomes more pronounced with every adjustment and the gap-filled data suggest that the number of people affected peaked in the mid-20th century, with no significant trend thereafter. After all corrections are applied, a downward

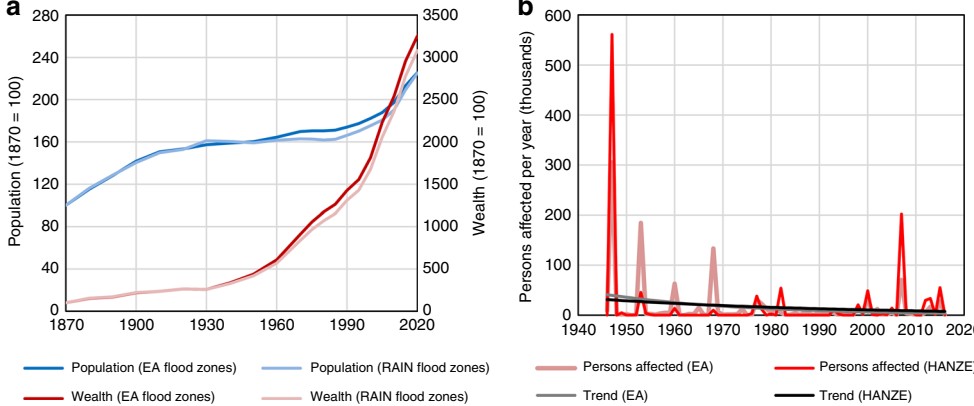

**Fig. 7** Validation of flood trends. **a** Trends in population and fixed assets living within 100-year flood hazard zone in England, using Environment Agency (EA) flood risk map and RAIN project map used in this study. **b** estimated persons affected (normalized) in England, compiled by intersecting EA historical flood outlines with HANZE-Exposure population grid, and compared with normalized reported persons affected from HANZE-Events. The trends were calculated using Poisson regression

trend in financial losses becomes apparent, although for losses normalized by wealth a mid-century peak is indicated. In total, we estimate that flooding affected 0.03% of European population per year on average between 1870 and 2016, and generated losses equal 0.08–0.09% of GDP (depending on normalization variant).

**Validation of flood footprints**. Another source of uncertainty is the delineation of flood 'footprints'. Here, we used 100-year flood hazard zones from pan-European modeling carried out in project RAIN, which correspond to the climate and physical geography of the 1971–2000 period. However, we acknowledge that not every flood in the database is a 100-year event, and that the 100-year floodplain boundaries do not remain stationary over time, given, for example, changes in climate, river geometry, urban development, or construction of hydraulic structures[52, 53]. But, because detailed, local flood hazard maps and recorded outlines for historical floods are not readily available for all locations in Europe, we use the 100-year floodplain as a proxy for floodplain extent and as a delineation of areas subject to high flood hazards. To validate the assumption that the 100-year is a viable proxy, we recalculated the results for England using flood extents from a comprehensive study by the Environment Agency (EA)[54]. Trends in exposure inside and outside the flood hazard zones are very similar for both pan-European maps from RAIN project and more detailed maps from EA (Fig. 7). The normalized number of affected persons within actual flood outlines recorded by EA yields an annual downward trend for 1946–2016 of 3.5%, compared to a 2% decline using the HANZE flood footprints and reported number of persons affected. However, the records are dominated by just a few events, especially the 1947 Thames valley flood and 2007 country-wide summer flood, hence there is large uncertainty in this comparison. The total (normalized) number of people within EA flood outlines for 1946–2016 is 1.11 million, compared to normalized reported number of people affected in HANZE of 1.19 million.

We also analyzed trends in reported annual losses for Poland between 1947 and 2006 based on national government statistics (Supplementary Fig. 9). For inflation-adjusted, but not normalized, losses an annual upward trend of 3.9% per year was found compared to a 4.2% increase in HANZE. Correcting for national GDP growth, reported annual losses still increase by 1.9%. In contrast, normalized and gap-filled data for Poland in HANZE indicate a 2.8% increase per year.

## Discussion

This study contains further sources of uncertainty which are less easily quantifiable. For instance, we assume that the flood hazard zones are constant over time. Climate change notwithstanding, many developments may alter local flood hazard, such as river regulation or construction of defences, bypass channels and reservoirs. In case of the latter, we include the erection of large reservoirs in land use, but do not consider their effects on the size of flood hazard zones. Other uncertainties are related to the normalization and gap-filling of damage statistics, though we include the probable margins of error in statistical significance testing (see Methods). Naturally, reported data could also contain many inaccuracies and inconsistencies. For example, there are many variations in the way that the number of people affected are reported across different sources, ranging from the number of evacuees to the number persons whose houses were either inundated or destroyed. Often, only the number of houses affected (flooded, damaged or destroyed) was provided for a given event. In this case, we assumed 4 persons per household, as some other national/international databases also used this assumption. In other cases, there might also be incomplete coverage of financial loss data, in the sense they do not always include all categories of assets. Information on area inundated more often than not refers only to agricultural land flooded rather than complete extent of events.

Nevertheless, the findings presented here are consistent with previous studies. No significant trend was reported for financial losses normalized at the country-level for major European floods (1970–2006)[14], major European windstorms (1970–2008)[29], or Spanish floods (insured losses, 1971–2008)[25]. For those time periods, insignificant downward trends were observed in the HANZE gap-filled financial losses normalized by wealth (−0.4 to −0.7% per year). In the United States, an insignificant annual decline of 0.49% was found in flood losses normalized by change in tangible wealth (1932–1997)[55]. This is similar to a 0.12% decline recorded in HANZE during those years for Europe. In Australia, no trend was found in insured losses from weather-related hazards for years 1967–2006, when the losses were corrected for increase in dwelling value[56]; however, in HANZE, an insignificant upward trend of 0.2% per year was observed.

Given the one-and-half century timespan of the study, an important question is raised as to whether the results indicate an influence of climate change. In the aforementioned study for the US, trends in precipitation were found to be similar to trends in

flood losses per capita. For Europe, we used the 20th Century Reanalysis[57] to obtain trends in the number of episodes of extreme precipitation above given return periods with a duration from 1 to 7 days, weighted by the size of flood zones within each grid cell of the reanalysis. An annual increase varied from 0.7–1.4% for a 10-year return period up to 0.8–2.4% for a 100-year return period. This is in between the value of increase both in the (unadjusted) number of flood events and (gap-filled) area inundated, which is contained in 1.4–1.7% range (for all floods, flash floods and river floods alike). In the Mediterranean region, there is a smaller increase, or even decrease in more recent decades, of extreme precipitation than in other parts of Europe which is also consistent with trends in number of events in the two sub-domains. The overall upward trend and the contrast between northern and southern Europe is consistent with other studies, both for extreme precipitation[13, 58, 59] and large flood occurrences[15, 16]. However, the number of events and flooded area must have had less pronounced trends for the continent as a whole, since the records of past floods have grown more complete over time, as shown in Fig. 6. This might indicate that, on average, flood hazard in Europe increased due to climate change and. Consequently, if the amount of losses mostly declined given constant exposure, vulnerability of population and assets decreased. On the other hand, given the significant deficiencies in the available data on flooded area, uncertainty in the under-reporting of smaller flood events and potential bias in reanalysis data, this relation could be coincidental. The average for Europe also masks large spatial diversity of meteorological and hydro-logical trends, let alone differences in adaption to flood risk. Also, in this study we did not consider localized pluvial floods, i.e., flash floods which occur in urban areas disconnected from riverine or coastal floodplains. Growing soil sealing by artificial surfaces connected with the aforementioned increase in frequency of severe (short and intense) rainfall events must have had an impact on the number of observed urban floods.

In future studies, research could focus on the influence of social, political and technical factors on changes in flood vul-nerability and risk. In this study, the most significant trend observed was a decline in flood-related fatalities of 1.4% per year since 1870 and 4.3% since 1950. Many technological factors could explain this decrease, such as vast improvements in commu-nication and transportation, which allowed more effective eva-cuation, rescue and relief operations, and the establishment of meteorological and hydrological agencies, which allowed for continuous observation and forecasting of rainfall and river dis-charges, improved early warning and disaster preparedness. Moreover, flood prevention, emergency management and disaster relief have largely become permanent government services, in contrast to ad-hoc local arrangements of the past. In general, vulnerability has declined compared to GDP per capita, as evi-denced in Supplementary Table 3 in which we analyzed rank correlation between relative losses (reported, but not gap-filled versus potential) for 310 major floods (the uppermost quintile in Fig. 5) and GDP per capita, finding all variables negatively cor-related. The strongest correlation was between GDP per capita and monetary value of losses, weakest with area inundated. This is somewhat similar to global-scale findings for modern coun-tries[31], but here we can trace those effects over the same sample of countries over almost one and half centuries.

Changes to the landscape could also have had an effect on vulnerability. Areas affected by floods urbanized to a higher degree than Europe in general (Supplementary Fig. 10), while croplands have been phased out faster. Dwellings, especially urban, have become sturdier as brick and concrete is more often used as construction material than timber or adobe. The per-centage of flood footprints under urban fabric has stronger

negative correlation with relative fatalities and people affected than GDP per capita for the 310 major floods mentioned in the previous paragraph (Supplementary Table 3). At the same time, there is a positive correlation between agriculture share of land and relative fatalities, persons affected and monetary losses. Analyzing the structure of wealth, the only types of wealth more strongly correlated with relative losses than GDP per capita is the share of infrastructure (for inundated area and persons affected) and agriculture (for fatalities and persons affected) in total wealth. This indicates that areas with high concentration of urban fabric and infrastructure are better protected than less important urban zones, let alone rural areas. This is an intuitive conclusion, but supported by evidence from events spanning almost 150 years. Further analysis may help to understand changes in flood pro-tection standards and land use-damage functions. Still, more data collection is needed, especially to gain a better understanding of local hydrologic trends. Only when the climate signal is fully removed from the data, can the trend in flood vulnerability be computed with confidence and the effectiveness of adaptation assessed.

## Methods

**General information.** The HANZE database, used as the basis for this study, includes records of damaging historical floods and a dataset of gridded land cover/ use, population, GDP and wealth that allows us to calculate changes in exposure within any given flood 'footprint'. HANZE covers 37 countries and territories in Europe: all 28 European Union member states, all four European Free Trade Agreement members (Iceland, Liechtenstein, Norway and Switzerland), four microstates located in Western Europe (Andorra, Monaco, San Marino and the Vatican) and the Isle of Man. The domain excludes the Canary Islands, Ceuta, Melilla, the Azores, Madeira and Northern Cyprus. Below, a summary of the methodology is presented. Further information about HANZE can be found in the database documentation[60].

**Modeling changes in exposure.** The general methodology is based on the con-cepts used to build the HYDE database[61, 62]. First, two detailed maps of population and land use are compiled for one point in time–'baseline maps'. Other time points in the past and in the future were calculated based on those baseline maps. Here, the maps refer to the year 2011/12, and have a spatial resolution of 100 m. For the years between 1870–2020 only the total population and land use at NUTS 3 regional level (1353 units)[63] is known. Hence, for each time step, the population and the different land use classes was redistributed inside each NUTS 3 region in order to match the regional totals.

Baseline land cover/use was taken from Corine Land Cover (CLC) 2012, version 18.5a[49] and population from GEOSTAT grid containing figures from 2011 population censuses[50]. The population grid was further refined to 100 m resolution using two disaggregation methodologies described by Batista e Silva et al.[64]. First, the 1 km population was redistributed into land use classes within each grid cell using an iterative 'limiting variable' method (M1 in the aforementioned paper) and CLC 2012 map. Then, population from land use classes was further distributed into 100 m cells proportional to soil sealing (method M3 in Batista e Silva et al.) taken from the Imperviousness 2012 dataset[65].

A database of statistics covering years 1870–2020 at NUTS 3 level was compiled from multiple sources covering population number, percent living in urban areas, persons per households, percent of land covered by croplands and pasture, and area covered by transportation infrastructure. The land use and population distribution was modified starting from the baseline map as follows. Change in population per urban grid cell was firstly considered proportional to the mean number of persons per household. Surplus population and urban fabric from this procedure was removed starting with grid cells furthest away from urban centres until it reach the total urban population per region recorded in historical statistics. Then, area covered by industry was changed proportionately to GDP per capita in the industrial sector, in constant prices. Grid cells located furthest from the urban centres were removed first, starting with the baseline map and moving in time from it. Reservoirs were removed when the year of the map was earlier than the year of an reservoir's construction, taken from GRanD database[66]. Grid cells of road and rail infrastructure were redistributed to match historical statistics on their total length per region, with grid cells reallocated as in case of industry. Airports were removed from the map when the year of the map was earlier than the year of an airport's construction. Airport land use class in CLC 2012 was connected with actual airports mostly through OurAirports database[67] and year of construction was found in Internet resources. Construction sites were removed for maps before 2010, and burnt areas for before 2005, otherwise were kept unchanged. Croplands were redistributed to match their historical area per region. Grid cells with the lowest value of suitability index for agriculture were removed first and grid cells not

used economically with the highest suitability index were added first. Suitability index for agriculture is proportional to slope (from EU-DEM[68]) and FAO crop suitability index for high-input cereals[69]. Pastures were computed in the same way as croplands, only with replacing the FAO crop suitability index for cereals with index for high-input alfalfa. In cases where land has become unoccupied after the application of the aforementioned procedure, natural land cover typical in the nearest neighborhood of a grid cell was assigned. If there was no natural land cover in the vicinity, forest land cover was assigned. Finally, the population of grid cells which transitioned from urban to non-urban during the calculation was reduced according to a value specific to each land use type. The non-urban population was changed proportionally to the evolution of mean number of persons per household. In case of further mismatch between output rural population and historical data per region, population was added/removed one person per grid cell at a time, starting with areas closest to urban centres. The remaining CLC 2012 classes (ports, dump sites, natural water bodies and courses, glaciers etc.) were assumed constant. Changes in urban population distribution was validated using a set of 42 population density cross-sections from 19 cities, spanning from 1871 to 1971. A reasonable match was achieved between reconstructed curves of population density-distance from city center relationship and estimates published in literature (see section 3.2.2. of HANZE database description[48] for more details).

As a last step, GDP (compiled at NUTS 3 level with sectoral breakdown) and wealth (non-financial, produced, tangible fixed assets compiled as a percentage of national GDP with sectoral breakdown and then multiplied by GDP at NUTS 3 level) were disaggregated to a 100 m grid. Fifty percent of GDP and wealth generated by agriculture (without forestry) was distributed proportionally among the population living in areas considered agricultural. The remaining 50% was uniformly distributed among agricultural CLC classes. Production and wealth in forestry sector was distributed as for agricultural, but using forest CLC cells instead of agricultural lands. Fifty percent of GDP and wealth in sectors of industry and services was distributed proportionally to the population of any land use class, and the other 50% uniformly distributed to industry or services-related CLC grid cells. The value of dwellings was distributed proportionally to the population of any land use class. The value of infrastructure was uniformly distributed to certain land use classes: roads, railways, airports, ports, and urban fabric.

**Compiling a database of flood events.** Records of flood events were gathered from many sources, ranging from news report and governmental data to publicly available databases and scientific literature. HANZE database covers the study area for years 1870–2016. Sources are identified per event in the dataset itself (see 'data availability'). Flood events, in order to be included in the database, had to fulfil certain criteria. First, information had to be available for at least one of the four damage statistics (area flooded, fatalities, persons affected and monetary value of losses). In case of fatalities equaling zero, data for any other variable had to be obtainable. Second, the minimum information required about the event was: country, regions affected by the event, year, month, type of event, cause of event. Insignificant floods, which affected only a small part of one region and had no fatalities, were not included in the database. Floods that were caused by insufficient drainage in disconnected urban areas, floods caused entirely by dam failure unrelated with a severe meteorological event, and floods caused by geophysical phenomena were also excluded. Events affecting more than one country were split in the database per country. Flood events were considered of 'compound type' if high river discharges or extreme precipitation occurred at the same time and location as high sea levels, and both have contributed to overall flood losses. Finally, the events were considered flash floods if rainfall that caused the flood lasted less than 24 h. However, urban floods (caused by inadequate capacity of the sewage system of a city), as well as floods of geotechnical (dam failure without an extreme hydrological event) and geophysical (e.g., tsunamis) nature were not included in the database.

**Flood footprints and normalization.** The extent of each flood event was obtained by intersecting a map of regions affected by an event with the flood map from the RAIN project[70], available from 4TU.ResearchData repository for river[71] and coastal[72] floods. The flood maps are for a 100-year return period and historical scenario (1971–2000). The floodplain includes all river sections with a catchment area above 100 km². The map does not include flood defences and therefore constitutes all potentially inundated areas. It should be noted that seven events were not included in the normalization and further analysis due to lack of flood extent data: four flash floods in Malta (where river were too small for inclusion in RAIN flood map) and three coastal floods in Sicily (where no flood risk was indicated in RAIN map).

Normalization was carried out by multiplying reported losses by the relative change in population, GDP or wealth within each event's footprint. As an example we can consider the 1953 North Sea flood in the Netherlands, which caused 1835 fatalities and 4.8 bln euro damages in 2011 prices. Given that the population within the flood's footprint increased by 60% and wealth by 636% between 1953 and 2011, the normalized fatalities will amount to 2930 persons and financial losses to 35.5 bln euro. It is therefore assumed that the vulnerability is constant within the timeframe of the study and all losses would have changed proportionally to local demographic and economic growth.

**Correcting for gaps in historical data availability.** Missing information on losses for events recorded in HANZE database was filled based on correlation between the four variables describing flood damages. Normalized values relative to potential damages within a given flood footprint were used. The empirical distribution of each variable was converted to ranks and the joint distribution of each pair of variables was fitted to five types of copulas (Gaussian, Gumbel, Clayton, Frank and Plackett)[73]. The best-fitting copula for each case was chosen according to the "Blanket Test" described by Genest et al.[74], which uses the Cramèr–von Mises statistic. For a given event and missing data, the available variable that was most highly correlated with the missing particular sample of the variable of interest was used. The conditional copula was sampled 10,000 times to generate samples of the conditional distribution of interest and mean of the conditional damage was used as the estimate of the missing values. The relative damage was the multiplied by total exposure within a given flood event's footprint. The graphs of dependency structures (transformed to standard normal space) are shown in Supplementary Fig. 4 with correlations and best-fitting copula types are included in Supplementary Table 1.

Underreporting of smaller flood events in the past was estimated by transforming normalized and gap-filled damage statistics (with financial losses normalized by wealth only) to ranks (highest to lowest) and dividing the events into quintiles based on their average rank. It was then assumed that the catalog of events in the upper quintile (20%), i.e., the most severe events, is complete over the entire dataset. For the other four quintiles, the catalog is assumed to be complete only during the most recent period: 1990–2016. During this period, the ratio of events between four lower quintiles to the highest one was 1.60, 2.02, 2.42 and 2.29 (higher quintile to lower). For other 30-year time periods (1870–1899, 1900–29, 1930–59, 1960–89) the ratio is lower, which was considered to be a function of underreporting of less severe floods (Supplementary Fig. 5). Hence, reported flood events were multiplied by factors necessary to achieve the same ratios between quintiles as in 1990–2016, where the highest quintile was not adjusted as we assume the records of most severe floods are complete. The same factors were applied to multiply flood consequences for all variables.

**Analyzing trends in flood risk.** Trends were analyzed using Poisson regression, which is better suited for count data than linear regression[75, 76]. Statistical significance of the trends presented in the paper was analyzed by Monte Carlo simulation. The trend calculated for a given variable (rate parameter of Poisson regression) was compared with 10,000 samples of randomized data series. Those randomized series were annual number of flood events or their consequences, where each flood event had a randomly assigned year from a uniformly distributed interval (1870, 2016). For each of the 10,000 randomized series the Poisson regression was calculated in order to obtain confidence intervals. The trend for a given variable was considered significant if the rate parameter was higher than in 95% of trends of randomized data series. As an additional check, the t test was applied to the calculated trends, yielding the same results at α = 0.05 significance level.

Reported values of variables were then 'normalized', i.e., for each flood footprint the reported value of losses were multiplied by the change in exposure between year of event and 2011 baseline. To test statistical significance in the normalized data series, we first estimated the uncertainty distribution of past exposure. It was assumed to be a log-normal distribution fitted to the empirical distribution of change in exposure between given time point and 2011 within all NUTS 3 regions. This log-normal exposure distribution was sampled to obtain a random value of exposure per given flood event. This sampling was repeated 10,000 times for each flood event to generate a set of randomized data series of annual normalized flood losses. This allowed us to compute uncertainty ranges in normalized data series in Supplementary Fig. 5. Then, a randomized data series were further randomized by assigning a year from a uniformly distributed interval (1870, 2016) to each flood event, as in previous paragraph. The trend was considered significant if it was higher than 95% of randomly generated trends.

For gap-filled data series, the uncertainty in the modeled data was further incorporated into significance testing. For each missing value of flood loss for a given event, 10,000 samples of marginal distribution of that variable obtained during the copula analysis. This allowed us to compute uncertainty ranges in normalized data series in Supplementary Fig. 6. Like for normalization, the data series incorporating uncertainty of gap-filling were further randomized by assigning a year from a uniformly distributed interval (1870, 2016) to each flood event, as in previous paragraph. The trend was considered significant if it was higher than 95% of randomly generated trends.

**Validation.** Comparison of exposure and flood losses trend was carried out using two Environment Agency (EA) maps. 'Risk of Flooding from Rivers and Sea', April 2017 version, contains highly detailed flood hazard zones at several probabilities of occurrence[54]. 'Recorded Flood Outlines', May 2017 version, contains actual flood extents continuously recorded since 1946, with a limited number of events from earlier years as well[77]. The flood hazard zones were intersected with population and wealth maps for 1870–2020, and the recorded outlines since 1946 were intersected with the disaggregated baseline population map. Additionally, we compared trends reported annual losses for Poland for 1947–2006 with the trends based on

HANZE-Events. Annual losses from Polish sources[78–80] were normalized using national GDP series.

Precipitation trends were computed using NOAA-CIRES 20th Century Reanalysis, version 2c[57]. It is a global climate reanalysis for 1851–2014 with a 3-h temporal resolution and 2 spatial resolution. A total of 329 grid cells intersect with the study area, for which daily precipitation amounts were extracted for years 1870–2014. For every grid cell an empirical return period (from 10 to 100 years) of 3, 6, 12-h and 1, 2, 3, 5, and 7-day precipitation was calculated and then the number of events which exceeded this threshold was obtained. Finally, this number of extreme events was weighted by the size of 100-year river flood hazard zones within each grid cell. Trends were also analyzed separately for Mediterranean countries (Cyprus, Greece, Italy, Malta, Spain and Portugal) and remaining countries in the domain. However, comparison of trends in the 20th Century Reanalysis with daily gridded observations from E-OBS[81] since 1950 shows potential bias in the reanalysis. In E-OBS, trends are quite uniform across time and duration of rainfall, in contrast to much larger variability in the reanalysis.

**Data availability.** The HANZE database used in this study is publicly available from 4TU.ResearchData with the identifier 'DOI: 10.4121/collection:HANZE' (ref.[47]) and from the corresponding author upon reasonable request.

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

## Acknowledgements

This work received support through the project "Bridging the Gap for Innovations in Disaster Resilience" (BRIGAID), which received funding from the European Union's Horizon 2020 research and innovation programme under grant agreement no. 700699. Further support was provided by the project "Risk Analysis of Infrastructure Networks in response to extreme weather" (RAIN), which received funding from the European Union's Seventh Framework Programme for research, technological development and demonstration under grant agreement no. 608166.

## Author contributions

D.P. conceived and designed the study, prepared and analyzed the data, and wrote the manuscript. A.S. assisted in the interpretation of the data, participated in technical discussions, and helped compose the manuscript. O.M.N. and S.N.J. helped guide the research through technical discussions. All authors revised the manuscript and gave final approval for publication.

## Additional information

**Competing interests:** The authors declare no competing interests.

