## [Peer Review File · Nature Communications]

Reviewers' comments:

Reviewer #1 (Remarks to the Author):

Dear Authors,

I have sincerely appreciated your paper. Due to the relevance of the subject, I hope it will be possible to improve it, reason for which I have tried to offer a contribution, writing my comments.

My comments are added as pdf file together with a figure.

Maurizio Polemio

The described scientific experience should be considered very interesting and of high scientific relevance. The paper main purpose is to define the flood damaging trend in Europe. The subject is extremely complex; the purpose is of high practical utility.

I wrote main suggestions and minor remarks, ordered page by page. Suggestions and remarks are described with the purpose to offer a contribution, I hope useful, to improve the quality of the paper. I added some references and a figure IPCC (2014, figure 3-6), some of these based on my personal research experience, supporting the content of main suggestions 2 and 3.

Main suggestions

- 1) Floods can be basically considered natural phenomena due to high or extreme rainfall; damaging floods are floods able to hit anthropogenic activities; the flood risk concerns the risk due to floods able to trigger damages (damaging flood and flood risk can exist only where and when some anthropogenic activity exists in the flooded area). Extreme rainfall, floods, damaging floods, and flood risks are cited and discussed in the paper but not with a sharp clearness. The term “event” is sometime used alone or together with flood: i.e. in the Figure 2 the title of the left figure is “floods events” but the caption concerns “events”. “Precipitation events”, “flood-prone areas”, “flood zones”, “extreme flood event”, “small event”, and “severity” are secondly used (when can a flood event/event be considered extreme/small?).
 - a. Please, add clear definitions of these terms from the beginning and use it in the rest of the paper, avoiding the use of synonyms as more as possible.
 - b. The paper mainly concerns damaging floods, from my point of view.
 - c. “flood risk” should be changed in the title using other words focusing on flood damage discussion which is at the core of the paper.
 - d. The case of a single transboundary flood is considered in more than a country: how the overestimation of the number of damaging floods (events) is avoided in the global discussion?
- 2) The whole Europe as unique spatial domain in which to discuss long-lasting temporal damaging flood variations doesn't seem an adequate choice as relevant inhomogeneous flood features exist, as shown by your dataset (Figure 2). In my opinion, the simplest spatial discussion should include two sub-domains.
 - a. The Mediterranean countries, which include Italy and Spain, corresponding to the first and the second peak of the national total value of floods, correspond to areas mainly hit by flash floods; the rest, with lower number of events, corresponds to a low number of very large river floods. The duration of triggering rainfalls, the forecasting difficulties, the mitigation criteria, and ... are completely different for these two types of damaging floods.
 - b. Recent results on measured rainfall and future climate scenarios seem to show the climate change is roughly dividing the Europe in two portions. In a portion of the Mediterranean area, including Southern Italy, there isn't evidence of increasing trend of extreme rainfall, in the rest this feature seems almost realistic and worrying.
 - c. Main focusing discussion and validation concern UK and Poland, both located in one of two (hypothetical) sub-domains.
- 3) The used geodatabase includes maps of land cover/use from 1870 but the role of land use modification isn't discussed and so is practically neglected. It seems unrealistic the land use modifications can't contribute to explain modifications of damaging flood feature trends.
 - a. The flood zones are considered constant over time in the paper also if the XX century was dominated by widespread hydraulic works in the whole Europe, able to overlap wide change to flood-prone areas and to the use of these areas.
 - b. It is suggested to add the discussion of the role of land use modifications or to explain in details the reasons for which this discussion wasn't realised/described.

Remarks

Where	Remark
Abstract, line 1	“...in time and is: please check.
Figure 1 and its discussion	The discussed period is 1870-2015 but the rest of the paper concerns the period 1870-2016. It is suggested to use a unique study period.
Discussion: “20 th Century Reanalysis”	Return periods of 5 and 10 years are considered but these values seem too low. It is commonly used a discussion referred to greater return periods, starting from 100 years. It is recommended to add the discussion of 100-year return period trend. The considered rainfall duration is greater than 1 day but the majority of considered damaging floods are flash floods, due very often to rainfall very shorter than 1 day. Please enlarge the discussion to consider this remark.

References

- IPPC, 2014, Climate Change 2014: Impacts, Adaptation, and Vulnerability. <http://www.ipcc.ch/report/ar5/wg2/>
- Lonigro, T., Gentile, F., and Polemio, M., 2015, The influence of climate variability and land use variations on the occurrence of landslide events (Subappennino Dauno, Southern Italy): *Rendiconti online Società Geologica Italiana*, v. 35, p. 192-195.
- Polemio, M., and Lonigro, T., 2015, Trends in climate, short-duration rainfall, and damaging hydrogeological events (Apulia, Southern Italy): *Natural Hazards*, v. 75, no. 1, p. 515-540.
- Petrucci, O., Pasqua, A. A., and Polemio, M., 2012, Flash Flood Occurrences Since the 17th Century in Steep Drainage Basins in Southern Italy: *Environmental Management*, v. 50, no. 5, p. 807-818.
- Polemio, M., and Petrucci, O., 2012, The occurrence of floods and the role of climate variations from 1880 in Calabria (Southern Italy): *Natural Hazards and Earth System Sciences*, v. 12, no. 1, p. 129-142.

(a)

(b)

Editorial Note: The figure on page 4 of this Peer Review File is reproduced with permission from IPCC (Figure 3.6, from Part A, Chapter 3: Freshwater Resources; IPCC, 2014, Climate Change 2014: Impacts, Adaptation, and Vulnerability).

Reviewer #2 (Remarks to the Author):

Dear authors, dear editor,

It was a pleasure to review this interesting and well-written manuscript. The authors clearly put a lot of thought and effort in this paper as well as the underlying HANZE database. Using a combination of modelling and recorded loss data, the paper presents, in my view, the best analysis on trends in European flood risk to date.

However, there are four significant issues that need to be addressed thoroughly before the paper can be published.

First, there are at least two recent global-scale studies on normalized trends in flood risk that arrive at very similar conclusions as the authors: Visser et al (2014; Climatic Change) and Jongman et al (2015; PNAS). The first applies EM-DAT and a large-scale normalization methodology; while the latter paper applies a similar methodology as the authors, namely a hydrological flood model and simplified historical exposure data in combination with MunichRe losses. These previous studies (perhaps there are others) use similar concepts and arrived at similar conclusions as the authors – they therefore reduce some of the claimed novelty and should at least be fully incorporated throughout.

Second, I am of the opinion that the authors overpromise on the use of the historical dataset. It is nice that the loss data covers a long time period going back to 1870. However, as the authors indicate themselves, the data before 1970 is extremely incomplete – this is the reason why all other trends studies decide to not go back before that period. For the early decades, the dataset used only covers large events with significant numbers of fatalities and misses all smaller floods with less impact. The entire dataset already has a significant bias as it is – with 36% of the data entries being in Italy, leaving only 1000 events for the other 36 countries – and this is especially problematic for the earlier period. As such, deriving any sorts of comparable trends from this period is highly questionable and leads to misinterpretations. The authors try to compensate this in the section on ‘underreporting’. This section is interesting and provides an estimate of the percentage of events that are not reported. However, this does not change the main estimates in the paper that were presented earlier (and which will make their way into the newspapers if this gets published). I would suggest the authors to critically review the dataset and define a start date from which they are relatively confident in presenting results, or perhaps split the dataset in two. Once again, it is nice to have a paper going back 150 years, but only if the data allows.

Third, there are issues with the use of the flood model. The authors dully describe that 56% of the reported floods in the database are flash floods. However, their hydrological model is suitable for fluvial floods is not able to actually represent flash floods (correct me if I'm wrong). Therefore, the relationship between the model and reported impacts is not correct. Either flash floods should be left out, or the authors should clearly defend that it is methodologically possible to include them. In addition, flood defenses are totally left out of the model and are only mentioned at the very end. Whereas I understand this will be complicated to do, I would suggest to integrate a discussion on changing protection standards over time as an explanatory factor throughout the results.

Fourth, the exposure model has several strong simplifications and is, as far as I can see, not validated. Changes in population are modelled using assumptions on urbanization by basically removing urban area from the fringes down to the center going back in time and redistributing population in the rural areas. This spatial 'trick' – which is necessary and also applied in other large-scale studies – is not validated using historical urban area maps. The resulting exposure distribution is thereafter overlaid with spatially explicit flood hazard maps and therefore totally determine the results presented in this paper. The paper needs a clear description and validation of the exposure data and an honest representation of the limitations. The claims made on the changes in population density inside/outside flood prone areas are not well defended and not explained. Perhaps the authors can look at using some historical maps for one or two case studies to indicate what the uncertainties are of the historical exposure data, and how it influences the results.

References

Visser, H., Petersen, A.C., Ligtvoet, W., 2014. On the relation between weather-related disaster impacts, vulnerability and climate change. *Clim. Change*. doi:10.1007/s10584-014-1179-z

Jongman, B., Winsemius, H.C., Aerts, J.C.J.H., Coughlan de Perez, E., van Aalst, M.K., Kron, W., Ward, P.J., 2015. Declining vulnerability to river floods and the global benefits of adaptation. *Proc. Natl. Acad. Sci. U. S. A.* 112, 201414439. doi:10.1073/pnas.1414439112

We would like to thank the referees for the time spent in reviewing our article and their valuable comments, which helped to improve our study considerably. Below, we list all the comments (in black font) and our responses (in blue font).

Reviewer #1 (Maurizio Polemio)

1. I added some references and a figure IPCC (2014, figure 3-6), some of these based on my personal research experience, supporting the content of main suggestions 2 and 3.
2. Floods can be basically considered natural phenomena due to high or extreme rainfall; damaging floods are floods able to hit anthropogenic activities; the flood risk concerns the risk due to floods able to trigger damages (damaging flood and flood risk can exist only where and when some anthropogenic activity exists in the flooded area). Extreme rainfall, floods, damaging floods, and flood risks are cited and discussed in the paper but not with a sharp clearness. The term “event” is sometime used alone or together with flood: i.e. in the Figure 2 the title of the left figure is “floods events” but the caption concerns “events”. “Precipitation events”, “flood-prone areas”, “flood zones”, “extreme flood event”, “small event”, and “severity” are secondly used (when can a flood event/event be considered extreme/small?).
 - a. Please, add clear definitions of these terms from the beginning and use it in the rest of the paper, avoiding the use of synonyms as more as possible.

We have revised the terminology throughout the paper to make it more consistent. In the introduction we revised the paragraph explaining the HANZE-Events database and added an additional explanation: “For the purpose of this analysis, ‘flood events’ (or simply ‘events’) refer only to damaging floods fulfilling criteria for inclusion in the HANZE database (see Methods for details). Also, when flood events are specified to be ‘small’ or ‘major’, the distinction pertains to severity of floods, i.e. the amount of losses generated by those floods relative to the overall distribution of losses for all events where small floods are those in the lower percentiles of this distribution and major floods are those in the upper percentiles.” We also clarified “flood zones” as “flood hazard zones”, as they refer to maps of flood hazard for a given return period.

- b. The paper mainly concerns damaging floods, from my point of view.

That is true, it only concerns damaging floods; to address this comment, a clarification was added in the introduction: “The second dataset (HANZE-Events) includes records of 1,564 damaging flood events that occurred within the same domain between 1870 and 2016, and had adverse consequences to people or property (damaging floods).” Also clarified in the sentence mentioned in the previous point.
 - c. “flood risk” should be changed in the title using other words focusing on flood damage discussion which is at the core of the paper.

To address this comment, we have changed the title to “Trends in flood losses in Europe over the past 150 years”.

- d. The case of a single transboundary flood is considered in more than a country: how the overestimation of the number of damaging floods (events) is avoided in the global discussion?

It is difficult to exactly calculate the number of cases due to changing political borders (a historical transboundary flood may not be such under present borders and vice versa), different dates of start/end of flood, or the fact that for each country, the data often came from various national sources, with various completeness of loss statistics. We estimate that the total number of events is elevated by approximately 2% by splitting events by countries.

3. The whole Europe as unique spatial domain in which to discuss long-lasting temporal damaging flood variations doesn't seem an adequate choice as relevant inhomogeneous flood features exist, as shown by your dataset (Figure 2). In my opinion, the simplest spatial discussion should include two sub-domains.

- a. The Mediterranean countries, which include Italy and Spain, corresponding to the first and the second peak of the national total value of floods, correspond to areas mainly hit by flash floods; the rest, with lower number of events, corresponds to a low number of very large river floods. The duration of triggering rainfalls, the forecasting difficulties, the mitigation criteria, and ... are completely different for these two types of damaging floods.

To address this comment, we have added an additional analysis in which we split the data into subdomains: Mediterranean countries (Cyprus, Greece, Italy, Malta, Portugal, Spain); non-Mediterranean countries. We also include the results per type of flood event: flash floods; river floods; river/coastal/compound floods. The results are described in a new section "Variation in flood loss trends by area and type of flood" and shown in Supplementary Table 2.

- b. Recent results on measured rainfall and future climate scenarios seem to show the climate change is roughly dividing the Europe in two portions. In a portion of the Mediterranean area, including Southern Italy, there isn't evidence of increasing trend of extreme rainfall, in the rest this feature seems almost realistic and worrying.

In addition to analyzing the flood trends for the two subdomains described in the new section "Variation in flood loss trends by area and type of flood", we also redid the precipitation analysis for those areas to account for the differences in rainfall trends across Europe.

- c. Main focusing discussion and validation concern UK and Poland, both located in one of two (hypothetical) sub-domains.

For validation, we used two alternative data series on flood losses and hazard zones to compare against the results we find using HANZE. Unfortunately, we haven't identified any alternative datasets for southern Europe (national flood databases of Italy, Portugal and Spain have already been incorporated in the HANZE database) and thus cannot draw any conclusions about the validity of our results for this area. The closest comparison can be made with insured flood losses for Spain (1971–2008), but here we are only able to compare trends based on very different methodologies as presented in the Barredo et al. 2008 paper (we observe a decline of normalized and gap-filled losses

for Spain of 0.5-1% per year compared to 0.1-0.3% in Barredo et al. for the same years). As this result is of limited relevance we omitted it from the paper.

4. The used geodatabase includes maps of land cover/use from 1870 but the role of land use modification isn't discussed and so is practically neglected. It seems unrealistic the land use modifications can't contribute to explain modifications of damaging flood feature trends.
 - a. The flood zones are considered constant over time in the paper also if the XX century was dominated by widespread hydraulic works in the whole Europe, able to overlap wide change to flood-prone areas and to the use of these areas.

It is true that flood prone areas might have changed due to new flood protection measures, but we consider here flood zones without flood defences in order to obtain information on actual losses contrasted with potential losses. Hence, improvements in flood control should be observed in the data as decline in vulnerability (expressed as monetary losses, deaths or persons affected). Without flood protection considered, the sensitivity of flood zones to different discharge scenarios is usually very low. We added an analysis on the correlation between land use/wealth structure and vulnerability, resulting in an intuitive conclusion that densely populated, urbanized areas with a lot of infrastructure are better protected than "less important" urban zones or rural areas.
 - b. It is suggested to add the discussion of the role of land use modifications or to explain in details the reasons for which this discussion wasn't realised/described.

We analysed the correlation between land use and vulnerability and added the results to the discussion and Supplementary Table 3. We also added Supplementary Fig. 10, which shows the trends in land use structure affected each year. Stronger urbanization and more rapid shift from agriculture to natural landscapes was observed within the flood footprints than for Europe in general.
5. Remarks
 - a. Abstract, line 1: "...in time and is: please check.

The introductory sentence was changed to: "Adverse consequences of floods change in time and are influenced by both natural and socio-economic trends and interactions."
 - b. Figure 1 and its discussion: The discussed period is 1870-2015 but the rest of the paper concerns the period 1870-2016. It is suggested to use a unique study period.

The figure and text was amended and now relate to 2016 statistics instead of 2015.
 - c. Discussion: "20th Century Reanalysis". Return periods of 5 and 10 years are considered but these values seem too low. It is commonly used a discussion referred to greater return periods, starting from 100 years. It is recommended to add the discussion of 100-year return period trend. The considered rainfall duration is greater than 1 day but the majority of considered damaging floods are flash floods, due very often to rainfall very shorter than 1 day. Please enlarge the discussion to consider this remark.

We have revised our precipitation analysis, by analyzing also 20-, 50- and 100-year return period, so the text now refers to the range of precipitation trends for 10- and 100-year return period. We analysed also sub-daily (3-, 6- and 12-hour) precipitation; trends were similar to 1-day precipitation, but the coarse grid of the reanalysis limits the utility of those results. Further, instead of combining extreme precipitation events equally from all grid cells, we now weight them according to 100-year flood hazard

zones to closer align the precipitation analysis to the remainder of the study. Finally, we checked the accuracy of the 20th Century Reanalysis by comparing it to E-OBS. There are large differences between those two datasets (the latter spanning back only to 1950), which reduces the confidence in the precipitation trends since 1870.

References

IPPC, 2014, Climate Change 2014: Impacts, Adaptation, and Vulnerability.

<http://www.ipcc.ch/report/ar5/wg2/>

Lonigro, T., Gentile, F., and Polemio, M., 2015, The influence of climate variability and land use variations on the occurrence of landslide events (Subappennino Dauno, Southern Italy): Rendiconti online Società Geologica Italiana, v. 35, p. 192-195.

Polemio, M., and Lonigro, T., 2015, Trends in climate, short-duration rainfall, and damaging hydrogeological events (Apulia, Southern Italy): Natural Hazards, v. 75, no. 1, p. 515-540.

Petrucci, O., Pasqua, A. A., and Polemio, M., 2012, Flash Flood Occurrences Since the 17th Century in Steep Drainage Basins in Southern Italy: Environmental Management, v. 50, no. 5, p. 807-818.

Polemio, M., and Petrucci, O., 2012, The occurrence of floods and the role of climate variations from 1880 in Calabria (Southern Italy): Natural Hazards and Earth System Sciences, v. 12, no. 1, p. 129-142.

Reviewer #2:

1. First, there are at least two recent global-scale studies on normalized trends in flood risk that arrive at very similar conclusions as the authors: Visser et al (2014; Climatic Change) and Jongman et al (2015; PNAS). The first applies EM-DAT and a large-scale normalization methodology; while the latter paper applies a similar methodology as the authors, namely a hydrological flood model and simplified historical exposure data in combination with MunichRe losses. These previous studies (perhaps there are others) use similar concepts and arrived at similar conclusions as the authors – they therefore reduce some of the claimed novelty and should at least be fully incorporated throughout.

We added the suggested references to the text, as they strengthen the case of our paper. The cited papers analyse flood losses only since 1980. Such a short timeframe, and the use of a single source of loss data is a big contrast to this study. Only 23% of events in our compilation are found in EM-DAT, and the vast majority of those that were included in EM-DAT were supplemented with additional sources of information on loss data before being included in the HANZE database. Furthermore, the publicly available information from Munich Re's NatCatService provided extra information for only 5% of the events in the HANZE database.

The previous studies cited above look at countries presently at very different levels of income, in contrast to this study, where the same group of countries is tracked from pre-industrial to post-

industrial era. We added this point to the discussion, where we now compare the GDP per capita with relative losses (vulnerability).

2. Second, I am of the opinion that the authors overpromise on the use of the historical dataset. It is nice that the loss data covers a long time period going back to 1870. However, as the authors indicate themselves, the data before 1970 is extremely incomplete – this is the reason why all other trends studies decide to not go back before that period. For the early decades, the dataset used only covers large events with significant numbers of fatalities and misses all smaller floods with less impact. The entire dataset already has a significant bias as it is – with 36% of the data entries being in Italy, leaving only 1000 events for the other 36 countries – and this is especially problematic for the earlier period. As such, deriving any sorts of comparable trends from this period is highly questionable and leads to misinterpretations. The authors try to compensate this in the section on ‘underreporting’. This section is interesting and provides an estimate of the percentage of events that are not reported. However, this does not change the main estimates in the paper that were presented earlier (and which will make their way into the newspapers if this gets published). I would suggest the authors to critically review the dataset and define a start date from which they are relatively confident in presenting results, or perhaps split the dataset in two. Once again, it is nice to have a paper going back 150 years, but only if the data allows.

We are aware of the limitations of the data and we have made an effort to clarify in the current version. For this reason we discuss five different starting points of the analysis – 1870, 1900, 1930, 1950 and 1970. At the same time, it is difficult to determine the minimum number of events adequate for confident conclusions. Indeed, Italy accounts for 36% of events in the database thanks, in part, to a superior national flood database, but those include, e.g., only 12% of all reported affected persons in the database. As noted above, the HANZE database goes far beyond EM-DAT or Munich Re compilations: the number of pre-1970 events included is almost twice the number of those included in EM-DAT post-1970. The analysis of underreporting of loss data in known events, and estimates of unreported events is an important contribution of this study; in contrast, loss data from international databases tend to be taken at face value in other studies.

3. Third, there are issues with the use of the flood model. The authors dully describe that 56% of the reported floods in the database are flash floods. However, their hydrological model is suitable for fluvial floods is not able to actually represent flash floods (correct me if I'm wrong). Therefore, the relationship between the model and reported impacts is not correct. Either flash floods should be left out, or the authors should clearly defend that it is methodologically possible to include them. In addition, flood defenses are totally left out of the model and are only mentioned at the very end. Whereas I understand this will be complicated to do, I would suggest to integrate a discussion on changing protection standards over time as an explanatory factor throughout the results.

The flood hazard zones were calculated for rivers with catchment area of at least 100 km². It is the most detailed flood map available for the whole study area, though it indeed does not represent floods in the smallest catchments. Nevertheless, the map still provided flood extents for all flash flood events considered except for Malta. We believe methodologically using a threshold of 24 h rainfall for dividing events into flash and river floods is adequate for the

purpose of this study; we also did not include urban floods not connected with any river system. We added this information to the methodology section.

However, to test the validity of the RAIN hazard maps, we made an additional calculation of normalized losses from flash floods where instead of using flood hazard zones we assume that the entire area of affected NUTS3 region constituted a potential flood zone. The results were added to Supplementary Table 2. The trends are very similar, with slightly larger decline in fatalities, less pronounced increase in persons affected and almost the same trend in losses except for the most recent period (from 1970). This indicates that demographic and economic growth was slightly lower inside our flood zones than in the entire affected regions (which can be explained e.g. by different rates of socio-economic growth in countries with different relative flood risk).

4. Fourth, the exposure model has several strong simplifications and is, as far as I can see, not validated. Changes in population are modelled using assumptions on urbanization by basically removing urban area from the fringes down to the center going back in time and redistributing population in the rural areas. This spatial 'trick' – which is necessary and also applied in other large-scale studies – is not validated using historical urban area maps. The resulting exposure distribution is thereafter overlaid with spatially explicit flood hazard maps and therefore totally determine the results presented in this paper. The paper needs a clear description and validation of the exposure data and an honest representation of the limitations. The claims made on the changes in population density inside/outside flood prone areas are not well defended and not explained. Perhaps the authors can look at using some historical maps for one or two case studies to indicate what the uncertainties are of the historical exposure data, and how it influences the results.

To keep the methods section concise we omitted the information on validation, which was actually carried out. We now added the following to the methods: “Changes in urban population distribution was validated using a set of 42 population density cross-sections from 19 cities, spanning from 1871 to 1971. A reasonable match was achieved between reconstructed curves of population density-distance from city centre relationship and estimates published in literature (see section 3.2.2. of HANZE database description [citation] for more details)”. Our other paper (<https://www.earth-syst-sci-data-discuss.net/essd-2017-105/>) contains the necessary details about the methods and validation of the urban population distribution. Currently, that paper is undergoing revision and will include a new European-wide analysis of the population grid utilizing municipal-level population figures from 1960 to 2010. It will be included in the link in the “discussion” tab, in the author’s response to reviewers and provide an statistics of match between modelled and actual municipal population for almost all of 1353 regions covered by this study.

Additionally, we included uncertainty distributions of past exposure (population, wealth, GDP) in testing significance of trends for normalized and gap-filled data. Also, the uncertainty ranges of trends in Supplementary Figures 5 and 6 include the uncertainty in past exposure.

References

Visser, H., Petersen, A.C., Ligtvoet., W., 2014. On the relation between weather-related disaster impacts, vulnerability and climate change. *Clim. Change.* doi:10.1007/s10584-014-1179-z

Jongman, B., Winsemius, H.C., Aerts, J.C.J.H., Coughlan de Perez, E., van Aalst, M.K., Kron, W., Ward, P.J., 2015. Declining vulnerability to river floods and the global benefits of adaptation. *Proc. Natl. Acad. Sci. U. S. A.* 112, 201414439. doi:10.1073/pnas.1414439112

REVIEWERS' COMMENTS:

Reviewer #1 (Remarks to the Author):

The authors considered all suggestions, modifying the draft. I have written two secondary remarks in the proposed docx file.

Best regards

Maurizio Polemio

Editorial Note: In their review of the second version of this manuscript, Reviewer 1 added their comments to the manuscript file. These comments, excluding minor textual revisions, have been copied into this Peer Review File.

On the third paragraph of the Discussions section:

Reviewer 1 Comment 1: *It is strongly suggested to compare these results with results of other studies, for the whole Europe, the Mediterranean area and subzones.*

Reviewer 1 Comment 2: *In the previous sentences, observations are describer former for the Europe and latter for the Mediterranean area. I am not sure if these sentences concern the former or the latter: please clarify or differ the discussion for both; please check but it seems the meaning could be better explained (rephrase)*

Reviewer #2 (Remarks to the Author):

I once again had the chance to review the revised manuscript. The revisions that were made have significantly improved the paper and addressed my concerns.

We would like thank the referees and the editor for the time spent in reviewing our revised article and their valuable comments. Below, we list the reviewers' and editor's comments (in black font) and our responses (in blue font).

Reviewer #1 (Maurizio Polemio)

1. It is strongly suggested to compare these results with results of other studies, for the whole Europe, the Mediterranean area and subzones.

We have reviewed existing literature and we found that they confirm the trends in our calculations, therefore we added the following:

“The overall upward trend and the contrast between northern and southern Europe is consistent with other studies, both for extreme precipitation [13,58,59] and large flood occurrences [15,16].

2. In the previous sentences, observations are describer former for the Europe and latter for the Mediterranean area.

I am not sure if these sentences concern the former or the latter: please clarify or differ the discussion for both; please check but it seems the meaning could be better explained (rephrase)
It was indeed unclear how the sentences relate to previous ones, therefore we have clarified it as follows:

“However, when considering bias in reporting, the number of events and flooded area must have had less pronounced trends for the continent as a whole. This might indicate that, on average, flood hazard in Europe increased due to climate change and, as a result, vulnerability of population and assets decreased.”

Cited literature:

[13] Moberg, A. et al. Indices for daily temperature and precipitation extremes in Europe analysed for the period 1901–2000. *J. Geophys. Res.* 111, D22106 (2006).

[15] Hall, J. et al. Understanding flood regime changes in Europe: a state-of-the-art assessment. *Hydrol. Earth Syst. Sci.* 18, 2735–2772 (2014).

[16] Benito, G., Brázdil, R., Herget, J. & Machado, M. J. Quantitative historical hydrology in Europe. *Hydrol. Earth Syst. Sci.* 19, 3517–3539 (2015).

[58] Klein Tank, A. M. G. & Können, G. P. Trends in Indices of Daily Temperature and Precipitation Extremes in Europe, 1946–99. *J. Clim.* 16, 3665–3680 (2003).

[59] Madsen, H., Lawrence, D., Lang, M., Martinkova, M. & Kjeldsen, T. R. Review of trend analysis and climate change projections of extreme precipitation and floods in Europe. *J. Hydrol.* 519, 3634–3650 (2014).

Reviewer #2

There were no further comments by the reviewer.